# Three-Dimensional Bioprinting for Retinal Tissue Engineering

**DOI:** 10.3390/biomimetics9120733

**Published:** 2024-12-01

**Authors:** Kevin Y. Wu, Rahma Osman, Natalie Kearn, Ananda Kalevar

**Affiliations:** 1Department of Surgery, Division of Ophthalmology, University of Sherbrooke, Sherbrooke, QC J1G 2E8, Canada; 2Department of Medicine, School of Medicine, Queen’s University, Kingston, ON K7L 3N6, Canada

**Keywords:** 3D bioprinting, biomimicry, biomimetics, retinal tissue engineering, regenerative medicine, bioinks, retinal cells, tissue scaffolds, regenerative medicine, microfluidics, retinal disease models, organ-on-a-chip

## Abstract

Three-dimensional bioprinting (3DP) is transforming the field of regenerative medicine by enabling the precise fabrication of complex tissues, including the retina, a highly specialized and anatomically complex tissue. This review provides an overview of 3DP’s principles, its multi-step process, and various bioprinting techniques, such as extrusion-, droplet-, and laser-based methods. Within the scope of biomimicry and biomimetics, emphasis is placed on how 3DP potentially enables the recreation of the retina’s natural cellular environment, structural complexity, and biomechanical properties. Focusing on retinal tissue engineering, we discuss the unique challenges posed by the retina’s layered structure, vascularization needs, and the complex interplay between its numerous cell types. Emphasis is placed on recent advancements in bioink formulations, designed to emulate retinal characteristics and improve cell viability, printability, and mechanical stability. In-depth analyses of bioinks, scaffold materials, and emerging technologies, such as microfluidics and organ-on-a-chip, highlight the potential of bioprinted models to replicate retinal disease states, facilitating drug development and testing. While challenges remain in achieving clinical translation—particularly in immune compatibility and long-term integration—continued innovations in bioinks and scaffolding are paving the way toward functional retinal constructs. We conclude with insights into future research directions, aiming to refine 3DP for personalized therapies and transformative applications in vision restoration.

## 1. Introduction

Three-dimensional bioprinting (3DP) is reshaping regenerative medicine, particularly for complex tissues like the retina. Initially developed for manufacturing, 3DP has rapidly advanced in healthcare, allowing the precise, layer-by-layer assembly of cells and biomaterials for functional tissue engineering. This review explores the fundamentals of 3DP, including its pre-processing, processing, and post-processing stages, as well as the primary bioprinting techniques—extrusion-, droplet-, and laser-based—which offer unique strengths but also present challenges for retinal tissue engineering [1].

The retina’s intricate structure, with its layered architecture and diverse cell types, presents unique challenges for bioprinting. Meeting these demands requires specialized bioinks designed for biocompatibility, printability, and mechanical alignment with retinal tissue. Recent developments in bioink formulations, such as hydrogels, are helping replicate the retina’s cellular environment, fostering cell viability and integration [2].

Despite promising progress, bioprinting viable retinal tissue involves hurdles, from maintaining cell health to ensuring functional tissue integration. Specifically, progress is limited by the lack of long-term viability and the underdevelopment of vascular structures necessary for sustaining the complex in vivo retinal environment. Advances in scaffold design, vascularization, and disease modeling are making strides toward biomimetic constructs suitable for drug testing and personalized therapies. Looking ahead, this review addresses the path toward clinical applications, focusing on overcoming immunogenicity, ensuring durability, and refining biomimetic strategies for potential breakthroughs in vision restoration.

## 2. Fundamentals of 3D Bioprinting

### 2.1. Definition and Principles

Three-dimensional bioprinting (3DP), also known as additive manufacturing, refers to a sophisticated printing process that employs computer-aided design (CAD) to facilitate a layer-by-layer deposition of living cells and biomaterials [1,3,4,5]. The concept of 3D printing was first introduced in 1986 with Charles Hull’s invention of stereolithography, creating solid objects from digital designs using light-cured resins [1,3]. While initial applications were concentrated in the sectors of business and manufacturing, the past two decades have witnessed significant advances in healthcare applications, including tissue engineering, regenerative medicine, and drug discovery [4,6].

Three-dimensional bioprinting is capable of synthesizing porous structures with highly controlled architecture, supporting the co-culture of multiple cell types, and allowing integration of vascularization in engineered tissues [1,3,4,5,6]. Unlike traditional fabrication techniques (e.g., electrospinning or decellularization), 3DP boasts greater precision, high throughput, and reproducibility for the purpose of replicating functional tissues [7,8]. Furthermore, the scalability of 3D bioprinting offers promise for future mass production of complex tissue constructs, including retinal tissue [1,4,5,6].

Emerging studies have begun leveraging machine learning and artificial intelligence (AI) to optimize printing parameters, predict cell behavior, and improve the fidelity of bioprinted tissues [9]. This advancement is promising for retinal tissue engineering given its lack of regeneration ability and the burden of irreversible retinal disease on millions across the globe [10,11,12].

### 2.2. Bioprinting Process Stages

There are three main stages of 3DP: pre-processing, processing, and post-processing [13].

(1) Pre-Processing: In this initial stage, bioinks, cells, and CAD designs are prepared [14,15,16]. Bioinks, typically comprising hydrogels, extracellular matrix components, or other cell-specific materials, are carefully selected based on the intended application [14,15,16]. Cells are harvested, cultured, and incorporated into bioinks [14,15,16]. CAD schematics, generated from patient-specific imaging data or pre-defined structures, provide the blueprint for the bioprinter [14,15,16].

(2) Processing: Here, printing occurs via the deposition and organization of materials [14,15,16]. Various parameters such as nozzle diameter, extrusion speed, and layer height can be adjusted as needed to optimize structural integrity and cell viability [14,15,16].

(3) Post-Processing: Finally, post-processing permits the final maturation of printed structures [17,18]. Depending on their intended function (e.g., transplantation, drug testing, or even to understand pathophysiology), cells now have a chance to fully integrate into their intended environment, differentiate, and interact harmoniously with native tissue [17,18]. Important considerations throughout the 3DP process include estimating and optimizing time usage and financial burden [17,18].

### 2.3. Types of 3D Bioprinting

There are three main types of 3DP—extrusion-based, droplet-based, and laser-based (refer to Figure 1) [16]. Extrusion-based 3DP involves a continuous stream of dispensation through a nozzle to form structures, such as in fused deposition modeling or bioplotting [16]. As the name implies, droplet-based involves droplets of bioink; in this category, you find inkjet bioprinting or microvalve based bioprinting. Finally, laser-based 3DP applies a laser or light source to cure a photosensitive resin layer by layer to build a final structure: stereolithography is an example of this type [16]. Each method possesses unique advantages and disadvantages related to printing capabilities, resolution, speed, scalability, material compatibility, ease of use, and commercial availability (see Table 1).

Droplet-based bioprinting emerged as the first 3DP technology in 2003 [21]. It operates similarly to 2D inkjet printing and is known for its high throughput, high resolution, cost-effectiveness, and high cell viability, making it suitable for creating complex multicellular constructs [19,20]. However, it requires bioink with an appropriately low viscosity to be ejected from the nozzle. This is in contrast to laser-based 3DP bioprinting, which employs a non-contact method that facilitates higher cell viability and compatibility with a wider variety of bioinks [19,20]. Previously, it has been effective in creating cell-laden constructs for tissue regeneration but is limited by high costs and system complexity [22]. Extrusion-based bioprinting, on the other hand, supports large-scale regenerative applications such as human-scale bone tissue [19,20]. In the context of retinal tissue, generally, extrusion-based methods are not commonly used due to the potential for shear stress on the delicate retinal tissues [23]. While there is a paucity of data on which retinal cells may be more susceptible to this damage, some studies emphasize that neural cells are particularly vulnerable, which might suggest similar sensitivity for retinal ganglion cells which are continuous with the optic nerve [24,25]. Laser-based and droplet-based techniques are commonly used to try to replicate retinal tissue [23]. However, the irradiation experienced by laser-based approaches may result in reduced cellular viability, while extrusion-based methods may cause damage to delicate retinal tissues because of the shear stress involved [23].

## 3. Retinal Anatomy and Challenges in Engineering Retinal Tissue

### 3.1. Anatomy and Function of the Retina

The retina is a highly specialized sensory region located in the back of the eye. It is both structurally and functionally complex, making it a formidable challenge for tissue engineering and bioprinting. It consists of several layers, with 60 distinct cell types (refer to Table 2) and an estimated total of 130 million cells, derived from the embryonic optic cup [3,8,9].

Retinal pigmented epithelial (RPE) layer: This layer, crucial for absorbing extraneous light and supporting photoreceptors, forms part of the blood–retina barrier, protecting photoreceptors from the vascular choroid. It regulates ion transport and vitamin A metabolism [6,7]. Importantly, the blood–retina barrier maintains retinal homeostasis protecting against pathogens and immune reactions [6,7]. These polygonal cells have complex intercellular junctions and extensive membrane invaginations (see Figure 2), reflecting their critical role in both metabolic support and maintaining the retina’s structural integrity [3,4,6,7]. The presence of melanin granules and various organelles involved in vitamin A processing, phagocytosis, and detoxification (e.g., peroxisomes) adds to the functional diversity of this layer. Additionally, this layer sits on Bruch’s membrane, a five-layered basement membrane and extracellular matrix between the retina and choroid (see Table 3 for acellular components found here). The RPE is also continuous with the choroido-capillary lamina of the choroid, which supplies the avascular RPE with oxygen and nutrients [4,5].

Neurosensory retina (NR) layer: Made of nine distinct layers, this layer contains photoreceptors, bipolar neurons, ganglion cells, and glial cells [26]. These structures coordinate to detect and convert light into neural signals, which are transmitted to the brain. The sheer complexity of these layers is illustrated by their distinct and vital functions (see Figure 3) [26]. Following the path of light, into the eye, these nine strata proceed as follows: inner limiting membrane, nerve fiber layer, ganglionic layer, inner plexiform layer, inner nuclear layer, outer plexiform layer, outer limiting layer, rod and cone layer (RCL), non-neural pigmented layers [26]. The RCL comprises 92 million highly light-sensitive rods (each with 600 to 1000 stacked flat membranous disks) and 4.6 million color-sensitive cones [26]. Importantly, these photoreceptors are non-motile primary cilia, and improper cilia orientation can have deleterious effects on vision [26].

Phototransduction—the process of converting light into electrical signals—occurs within the densely packed RCL, which contains billions of light-sensitive proteins (e.g., rhodopsin) [26]. With the exposure to light, various molecular and biochemical reactions are triggered; this intricate process repeats within milliseconds [26].

Furthermore, the retina contains several notable regions that contribute to its specialized functions (see Figure 4) [26]. The optic disk, colloquially known as the blind spot, is located in the retina where the axons from the nerve fiber layer converge and exit the eye as the optic nerve. This region lacks other NR layers [26]. The fovea centralis, directly opposite the pupil, facilitates maximal visual acuity and sharpness [26]. Surrounding the fovea is the macula lutea, where cone cells are rich in carotenoids, which filter potentially harmful short-wavelength light to protect the photoreceptors in the fovea [26]. Additionally, the retina contains nonvisual photoreceptors, a subset of ganglion cells located in the ganglionic layer [26]. These cells assist in regulating the human circadian rhythm [26].

### 3.2. Challenges in Replicating the Complex Structure and Function of Retinal Tissue

Replicating the complex structure and function of retinal tissue presents significant challenges due to its multilayered architecture, cellular diversity, and specialized physiological roles, making it one of the most difficult biological tissues to replicate [10]. The retina’s stratified layers, which include over 60 different cell types, must be recreated with precision, while ensuring proper vascularization and integration with the optic nerve [3,8,9,10]. Key obstacles include mimicking the blood–retina barrier and maintaining tissue stiffness within the physiological range of 10–20 kPa [15]. Additionally, scaffolds must support the intricate spatial arrangement and varying tissue properties [16]. Achieving a functional microvascular network is also critical, as it requires precise interactions between junction proteins, hormonal signaling, and cell polarity [17,18]. Finally, ensuring cell viability is difficult, with high rates of apoptosis posing a challenge to maintaining functional tissue [11,12].

Further, disease pathophysiology can pose an even greater challenge to reconstructing retina. Examples of conditions affecting the retina include age-related macular degeneration, diabetic retinopathy, pseudoxanthoma elasticum, and inherited retinal dystrophies.

## 4. Bioinks for Retinal Tissue Engineering

### 4.1. Requirements for Bioinks Specific to Retinal Applications

The 3DP bioinks for the development of retinal constructs have varying requirements depending on the retinal layer and desired replicated function. An example of this structure–function relationship can be seen in a two-layer retinal model of a photoreceptor cell layer deposited over a bioprinted Bruch’s membrane [28]. Masaeli and collaborators (2020) generated this bioink using a thin gelatin methacrylate (GelMa) layer, a material that contains matrix metalloproteinase peptide motifs, increasing existing cellular function to mimic the physiological microenvironment of the retina [28].

Bioinks in retinal tissue engineering are designed with a cellular component mimicking native retina and carrier material. Generally, carrier materials are either a hydrogel or a biomaterial laden with cells [29]. Hydrogels are three-dimensional cross-linked structures that, by name, can carry large volumes of water and are used in bioinks due to their similarity to the ECM and their suitability for cell growth [30]

A successful bioink can be assessed by the following three requirements: biocompatibility, printability, and display of similar mechanical properties to native retina (see Figure 5) [31]. Biocompatibility refers to the ability of a bioink to mimic the microenvironment of the printed cells in the retina to ensure normal cellular activity [29,32]. Printability assesses the rheological properties of the bioink (i.e., viscosity) to determine success in printing the construct [29,32]. Mechanical properties simultaneously assess the bioink’s ability to support cell culture and implantation into carrier structures. Their mechanistic role is six-fold and includes (i) mimicking the native extracellular matrix, (ii) cell viability and proliferation, (iii) promoting cell differentiation, (iv) facilitating retinal layer stratification, (v) improving retinal tissue integration, and (vi) enhancing cell viability under stress. Bioprinted structures are cultured in vitro, requiring cellular perfusion, and can result in degradation; the need for mechanical properties ensuring structural integrity is key [29,32]. The following case examples will review the retinal applications of these requirements to various available bioinks (refer to Table 4).

#### 4.1.1. Biocompatibility of Bioinks

Hydrogel carrier materials can be natural or synthetic. Successful physiological mimicry of the desired retinal microenvironment is highly dependent on the type of hydrogel used. Natural polymeric hydrogels include agarose, alginate, collagen, fibrin, gelatins, hyaluronic acid (HA), and Matrigel^TM^ [2,28,33,34]. Pluronic^®^ and polyethylene glycol (PEG) are two synthetic materials used in past retinal models [35,41]. Synthetic hydrogels often struggle with the promotion of cellular activities including adhesion and proliferation [2]. Specifically, PEG hydrogels are susceptible to photocrosslinking, which can affect cell viability post-printing [42]. Natural hydrogels have varying levels of cell viability and cellular growth encouragement, which comes at the cost of their printability and mechanical strength [2].

Cellular viability and promotion of differentiation is a major concern when testing bioink biocompatibility. Use of hyaluronic acid with methacrylation by a glycidyl–hydroxyl reaction, resulting in a photopolymerizable hydrogel (HA-GM), is a process applied in retinal tissue engineering to ensure cellular viability and differentiation post-printing [35]. In a hydrogel testing study, Wang and collaborators (2018) suggested that the adjustment of the degree of methacrylation in hydrogels and codifferentiation of cell types (specifically fetal progenitor cells and RPEs) in a bioink can improve long-term cellular viability and promote increased cellular differentiation in a retinal construct [35].

Bioinks should also be non-cytotoxic to the surrounding microenvironment. Belgio et al. (2024) developed an optimized sodium alginate-gelatin (SA-G) hydrogel to replicate the native photoreceptor layer to reduce cytotoxicity when applied in vivo [32]. When tested in vitro using a murine fibroblast cell line (L929, Catalog No. CCL-1^TM^, American Type Culture Collection, Manassas, VA, USA), cell viability was found to be higher than 90%, suggesting the novel bioink was non-cytotoxic and ideal for 3DP replication of photoreceptor tissue [32].

In cases where a hydrogel is not used, secondary cell types can be used as substrates to promote cell growth and ensure viability. To use inkjet 3DP to print cells of the adult rat retina, a bioink combining adult rat retinal ganglion cells (RGCs) and glia supported RGC viability via the growth-promoting properties of rat glial cells which were retained post-printing [43]. Decellularized ECM (dECM) is a common biomaterial used in place of hydrogels (see Figure 6). The dECM-based bioinks use ECM from an animal source (i.e., bovine, porcine) and solubilize dECM in acidic solution to create a printable gel material. This process supports cell viability by reducing immune rejection after printing to the native tissue given the similarity of the dECM to the natural ECM structure present in the human retina [44].

The ECM is a key component of the retina and its replication of its properties are essential for biocompatible and functioning 3D retinal constructs. The ECM acts as a catch-all for various growth factors, enzymes, and molecules necessary for functioning [45]. Specific explorations into the ideal ECM structure for bioprinting are dependent on the retinal tissue replicated, as the ECM components differ between layers; for example, the interphotoreceptor matrix (IPM) is a specialized ECM situated between the photoreceptor and RPE layers [46]. Unlike other ECMs found in the retina, the IPM lacks a collagenous structure and does not have laminin and fibronectin as major components. Instead, the structural components of the IPM are secreted by photoreceptors and RPE cells.

#### 4.1.2. Printability of Bioinks

The printability of a bioink is privy to several rheological properties, including viscosity, viscoelasticity, elastic recovery, and shear stress [32,36]. This is a prerequisite for successful 3DP, as these properties determine the preservation of the bioinks cellular components during the damaging process of bioprinting [32,36]. These properties also directly interact with one another, making optimization of a bioink complicated. For example, increasing the viscosity of a bioink can result in a higher shape fidelity during printing; however, this has the adverse effect of increasing shear stress which can damage cells in the bioink [36].

Shear stress is reduced through shear-thinning, a process of decreasing viscosity of a bioink to preserve cellular components in a bioink. For hydrogels, reducing shear stress can be achieved by adjusting temperatures and cross-linking. Temperature is inversely related to a hydrogel’s viscosity, and the ideal temperature for printability is determined by (1) the type of hydrogel and (2) the temperature at which the cellular components of the bioink are viable [37,38]. Cross-linking is a post-3DP procedure that makes changes to the printed hydrogel to achieve the ideal mechanical properties of printed tissue [38].

Viscoelasticity is a rheological property that considers a bioink’s ability to have viscous flow and maintain elastic shape retention [36]. Yield stress, the stress a bioink must overcome for a deformation to occur, is important to elastic shape retention. An increased yield stress in a bioink can make the final construct stronger and stiffer, however, it can also cause damage to cellular components [36]. Yield stress is generated in bioinks by additives, including gellan gum, hyaluronan, and carrageenan [36].

#### 4.1.3. Mechanical Properties of Bioinks

There are several mechanical properties in bioinks that determine the success of a printed construct. In retinal models, swelling degree (hydrogels) and cross-linking are topical properties that are optimized to induce success.

Hydrogels are composed of very hydrophilic polymer networks and swell up to 99% water (*w*/*w*) of their dry weight without dissolution [39]. Swelling degree ultimately determines the shape of the hydrogel and by extension the produced construct. Increasing hydrogel swelling capacity directly increases hydrogel stability [39]. Most hydrogels exhibit optimal swelling capacity at a physiological pH (~7.4) [37].

Cross-linking is a major mechanical property that is employed in retinal hydrogel-based bioinks to maintain structural integrity. Cross-linking can be physical or chemical. Physical cross-linking procedures are reversible and involve the manipulation of intermolecular relationships in the polymer-based hydrogel [37]. Chemical cross-linking is an irreversible process that ensures heightened structural stability, usually using temperature changes to induce chemical changes in the polymers [37]. Each hydrogel has an ideal temperature at which it can independently perform chemical cross-linking, though most occur at room temperature or around 37C [37,40].

Other mechanical properties that are of import in the general 3DP process include molecular weight, gelation kinetics, and stiffness [34].

### 4.2. Recent Advancements in Bioink Formulations

In the last few years, bioinks have rapidly developed to increase biocompatibility, printability, and mechanical properties.

A retinal decellularized ECM (RdECM) bioink developed from porcine retina has shown promising results for retinal tissue engineering [32]. The RdECM removed retinal cells from a porcine donor and mixed the RdECM base with 1% collagen to create a robust biomaterial [32]. The biomaterial also contained additives such as HA, heparin sulfate, and laminin, which are key ECM subcomponents in the human retina [32]. This bioink was further cultured with human Muller cells (MIO-M1), and the resulting construct exhibited differentiation of Muller cells [32]. Subsequent animal models using the RdECM construct demonstrated retinal protection ability, as mice treated with a laser that were implanted with the RdECM construct maintained the thickness of their retina when compared with untreated mice [47]. As previously discussed, the SA-G hydrogel bioink developed by Belgio et al. (2024) showed promising results in non-cytotoxicity and the potential to replicate the native photoreceptor layer in a printed construct [32]. This bioink further modifies a two-layer alginate hydrogel technology that proved successful (2017) for replicating RPE cells [41].

Bioink formulations have also been used in recent advancements to support disease modeling for common retinal pathologies. Wu et al. (2023) developed a bioink using 4% GelMa and 1% hyaluronic acid methacrylate (HAMA) and a cellular co-culture of human microglia (MG) and endothelial (EC) cell lines in a DLP bioprinting process for the modeling of diabetic retinopathy [48]. GelMA and HAMA have shown strong biocompatibility in retinal cell culture and are flexible in mechanical stiffness. The co-culture of MG and EC achieved cellular proliferation protection under high glucose conditions meant to mimic the glucose levels of diabetic retinopathy, while bioinks developed from MG or EC culture alone exhibited reduced proliferation at elevated glucose levels [48].

Printing processes greatly affect bioink considerations. Arman et al. (2023) created an experimental bioink for the laser induced forward transfer (LIFT)-based bioprinting system to study cellular viability in a mouse cone photoreceptor line [33]. Although laser printing systems avoid high shear stress found in extrusion-based bioprinting, the irradiative effects are a major concern for cellular viability [23]. This bioink consisted of collagen I and glycerol at varying concentrations mixed with a mouse cone photoreceptor line (661w) and successfully printed mouse cone cells in a droplet construct [33]. This work also emphasized the importance of environmental factors in bioink printing success and suggested that cellular viability could be increased by controlling environmental factors (dust, airflow, and temperature) [33].

## 5. Challenges in Bioink Preparation

### 5.1. Technical Challenges

Bioink materials should be robust enough to withstand the pressures of printing, ideal to support structural integrity post-printing, and should show significant cellular viability in culture for successful retinal 3DP [2,30,47,49]. Using decellularized animal retinal tissue for bioink biomaterial, as evidenced by Kim et al. (2021), provides a strong and biocompatible base [47]. Chemical modifications of hydrogels can also increase bioink strength by creating durable polymer networks and improving mechanical properties such as shear thinning and cross-linking [35]. The retinal microenvironment is structurally complex, making replication difficult as bioinks must achieve a fine enough resolution to address structurally complex pathways such as the photoreceptor conduction [31]. Appropriate vascularization of printed tissues remains difficult to achieve. The retina is a highly vascularized structure. It is supported by a capillary network that directly supplies the NR layer and a multiplex choroidal vasculature that maintains the outer retinal layers [50]. Given this vascular complexity, a tissue construct would necessitate integration into the existing retinal vascular bed to support oxygenation and nutrient supply of implanted tissue. Current bioprinters lack high-definition resolution to print the small vessels of the retinal vascular bed [51]. Masaeli et al. (2020) proposed that vascular replication could be supported during the printing process by releasing known angiogenic factors (VEGFs) from printed RPE layers [28]. However, the VEGF is a known carcinogen and reducing malignancy promotion long-term is a primary concern [52,53]. More models with a variety of known angiogenic factors in the retina would be required to consider this a viable solution in vivo.

### 5.2. Biological Challenges

The cellular diversity of the retina makes it difficult to print a fully functioning model of the entire retina. Past models have focused on generating scaffolds of one or two layers. To ensure cellular viability in these constructs, the cells used require strong regenerative capacities. Stem and progenitor cells used in current models include induced pluripotent cells (iPSCs), mesenchymal cells, and human fetal retinal progenitor cells [33,41,49,52,54]. At this stage, cell damage and injury are inevitable during the printing process, reducing viability. As mentioned previously, extrusion-based 3DP methods generate shear stress that physically damage cells [53]. Laser-based 3DP methods often generate thermal and radiative stressors that are chemically detrimental to cell growth and can induce apoptosis [54]. Mathematical models have been proposed to predict and reduce cellular damage in various models, however, these predictions are unlikely to be applicable in vivo [23,32].

Ensuring appropriate differentiation into functional retinal cell units that can functionally interact with the native retinal microenvironment is another major biological challenge. Improvements to scaffolding technology are supporting increased differentiation and longevity for retinal tissue engineering [55]. Hybrid scaffolding has been particularly useful to capitalize on the biological and mechanical advantages of multiple biomaterials simultaneously [56,57,58]. Long-term survival of cells when introduced to the native retina is another key challenge [55,59]. Currently, optimization models for biomimicry to specific retinal layers and cellular co-cultures in bioinks remain the strongest contenders for addressing cellular longevity in vivo [32,48].

## 6. Applications of 3D Bioprinting in Retinal Tissue Engineering

### 6.1. Overview of Current Research and Developments

Recent innovations in 3D bioprinting incorporate cellular biology, various biomaterials, and advanced printing methods to construct optimal functional retinal models, facilitating the study of diseases and development of therapies (refer to Table 5).

### 6.2. Advancements in Tissue and Cellular Engineering for Retinal Models

#### 6.2.1. Specific Cell Types

There are numerous studies that focused on generating and optimizing specific cell types in the retina. In 2014, Lorber et al. made the first significant attempt to 3D bioprint retinal tissue [60]. They focused specifically on printing retinal ganglion cells and glial cells using piezoelectric inkjet printing (a type of droplet-based 3DP) that generates pressure to eject small controlled droplets [60]. The survival rate, 69–78%, was comparable to controls and the glial cells retained their function [60]. At that time, their primary hurdle was sedimentation in the printing nozzle, limiting final cell count. Shi et al. (2018) bioprinted a type of RPE cell (ARPE-19) and photoreceptor cell (Y79 cells) using a microvalve-based technique [61]. They maintained viability and structure during the culture process [61]. However, similar to Lorber et al., sedimentation also impacted the cell count and precision of deposition [60,61]. Additionally, the complexity of cell patterning required additional steps to maintain fidelity [61].

Other cell types such as retinal glial cells (e.g., Muller cells) have also been attempted to be faithfully recreated. In 2023, Jung et al. compared 2D cultured versus 3D bioprinted Müller cells [62]. The latter was superior in replicating in vivo features (e.g., endfeet, soma, and microvilli) as well as mimicking physiological changes in diabetic conditions (via potassium and water channel expression, and cell cytokine and growth factors) [62].

Arman et al. (2023) demonstrated the ability to synthesize photoreceptor cells using laser-induced forward transfer (LIFT) bioprinting, discussed in length in Section 5 [32].

Beyond manufacturing cells, it is also important to consider their transfer. Lee et al. (2024) performed just that by exploring a novel method for contactless and damage-free extraction of RPE cells from a monolayer using acoustic droplet ejection [84]. This technique enables high-precision and contactless cell extraction, without damaging the cells.

#### 6.2.2. Optimizing Hydrogel Scaffolds

An appropriate scaffold has been a popular discourse within the existing literature. In the context of retinal tissue, hydrogel scaffolds provide structural support and mimic the extracellular matrix. Gellan gum (GG) is frequently chosen as the basis for hydrogels in retinal 3DP. Various studies have investigated which additives optimize the properties of GG hydrogels. These formulations are illustrated in Figure 7. The studies described below evaluate physicochemical properties (including but not limited to mass swelling, sol fraction study, a weight loss test, a viscosity test, an injection force study, a compression test, and a stress relaxation analysis) and functional properties (i.e., proliferation of RPE cells like ARPE-19) to determine the overall utility of the scaffold. For example, Kim et al. (2019) used a polyethylene glycol (PEG)/GG hydrogel with varying concentrations of PEG [63]. They found that the 3 wt% PEG/GG hydrogel showed superior biocompatibility (>90%), enhanced cell adhesion, and improved cell growth (of ARPE-19 cells) compared with pure GG hydrogel [63]. Soon after, Choi et al. experimented with eggshell membrane (ESM) and GG hydrogel [64]. At 4 w/v% concentration, the ESM reduced viscosity by 40%, reduced swelling by 30% due to its lower hydrophilicity, and increased degradation efficiency by 30% compared with pure GG hydrogels, though they did experience slightly weakened mechanical properties [64]. Next, Youn et al. (2022) investigated GG with hyaluronic acid hydrogel, demonstrating that it is a viable substrate for physicochemical and mechanical properties suitable for injection into the retina [65]. Kim et al. in 2022 used GG and silk sericin (SS) hydrogel [66]. A GG with SS 0.5% hydrogel had a compressive strength akin to natural RPE tissue (~10 kPa) and sufficient ARPE-19 cell proliferation [66].

An unconventional idea by Masaeli et al. (2020) involved creating a hydrogel-free alternative [67]. In this approach, an RPE layer was printed as living biopaper to position a fibroblast layer that can secrete its own supporting matrix [67]. The low shear stress applied during printing did not negatively affect cell survival, though viability was slightly lower than controls over a week post-printing [67].

Kim et al. in 2021 demonstrated that printing RPE onto polymer scaffolds alone is insufficient for full maturation [68]. Their Bruch’s membrane-mimetic derived from porcine bioink supported key RPE functions such as barrier integrity, clearance, anti-angiogenic factor secretion, and phototransduction, making it a promising scaffold for RPE transplantation [68]. Similarly, Masaeli et al. (2020) also explored a scaffold-free method using inkjet bioprinting depositing photoreceptors directly onto a bioprinted RPE layer [28]. They used a gelatin methacryloyl (GelMA) layer to mimic Bruch’s membrane. Results showed accurate positioning, expression of structural and functional markers (e.g., vascular endothelial growth factor) [28].

In regard to the method of printing for scaffolds, studies remain divided. Liu et al. in 2022 applied electrohydrodynamic jet printing to fabricate scaffolds that mimic Bruch’s membrane [69]. Both Worthington et al. (2017) and Shrestha et al. (2020) applied two-photon polymerization, a laser-based 3DP technique optimized for highly precise nanoscale strictures to create a strong scaffold to sufficiently support retinal progenitor cells and their growth/differentiation [54,70]. Wang et al. in 2018 also used laser-based 3DP to create a scaffold with adequate stiffness, decent cell viability (>70%), and successful support of fetal retinal progenitor cells [35].

#### 6.2.3. Integrating Multilayer Structures, Functions, and Vascularization

Shi et al. and Schecter both successfully created multilayered bioprinted retinal constructs, but Schecter’s three-layered model, including a choroid-like layer, offers a more comprehensive approach to treating retinal diseases like AMD [41,71]. Wang et al. introduced stem cell-derived RPE spheroids which enhance cell viability and maturity, adding regenerative potential to 3DP constructs [84]. Meanwhile, Cadle et al. addressed the need for more anatomically realistic vascularization by incorporating human ocular fundus images into the bioprinting process, a critical factor missing from the other studies [85]. Despite promising results, long-term viability, functional integration, and vascularization in these bioprinted retinas require further investigation and increased breadth in the literature for clinical application.

The functionality of 3D retinal bioprinted constructs is heavily dependent on the survival of cellular components, which can be improved with continued cell media optimization. Pollalis et al. (2024) demonstrated that miRNA found in extracellular vesicles (EVs) secreted by RPE cells have significant roles in supplementing RPE cell activity, further confirmed by a differentiation study with human embryonic stem cells [86]. Ensuring RPE secretion of EVs in the generation of RPE-based bioinks may support RPE survival and functionality in printed constructs. A barrier to optimization of RPE-specific cell media lies in the protocols used to generate the cells. Bharti et al. (2022) notes that there are no standard protocols to differentiate and propagate iPSC-derived RPE cells, further compounded by differences in substrate surface, plating density, feeding mediums and frequency, and duration of culture [87]. These differences were thoroughly considered in a recent optimization study for the culture of Muller cells for 3D bioprinting applications [88]. In this study, past protocols for the Muller cell culturing were compared for cell media effectiveness by experimenting with varying levels of growth factors and serum, generating a final protocol that was posited as ideal for Muller cell proliferation [88].

### 6.3. Retinal Disease Models

There is robust literature on 3D bioprinted models for simulation of retinal disease to understand underlying pathophysiology. Several retinal diseases present with RPE degradation. Kim et al. (2022) developed an in vitro model of RPE damage by exposing a bioprinted construct of RPE overlaying Bruch’s membrane to cigarette smoke [72]. The oxidative stress induced by the cigarette smoke was found to disrupt both RPE tight junction durability and the integrity of the outer blood retinal barrier (oBRB) [72]. The structure of the outer blood retinal barrier is further described in Figure 8. The disruption of the oBRB as a source of disease was further assessed by infection with the recombinant Zika virus in a 3D bioprinted oBRB model [73]. Dorjsuren et al.’s (2022) Zika model demonstrated a significant disruption in the blood–retina barrier function. The immunological diversity of the model described in this article is somewhat limited because it focuses on certain aspects of the host–virus interaction and barrier function without a full immunological response (e.g., cytokine release, chemokine gradients). Thus, this model is limited in its application to the general population, highlighting a need to develop immunological constructs that are applicable to a wider range of diseases [73]. Future models could be improved with greater fidelity to the natural ocular immune response, particularly through the role of the oBRB in the retina’s immune-privileged status and various resident immune cells (i.e., microglia, Müller cells etc.). Song et al. (2023) developed a model oBRB with a fully polarized RPE and choroidocapillaris network to simulate age-related macular degeneration (AMD) [74]. While this model is effective in testing new therapies for AMD, it is limited in accuracy for disease presentation as it lacks the melanocytes and innate immune cells present in the choroid [74]. In a recent study, Song et al. (2023) designed a more comprehensive, multilayered retinal model with the development of an oBRB tissue composed of endothelial cells, pericytes, and fibroblasts bioprinted on the basal side of a scaffold with an RPE monolayer on top [75]. The oBRB tissue acted as an excellent model for AMD in vitro. RPE degeneration in retinal dystrophy can be linked to excessive hydrogen peroxide (H202) production and subsequent oxidative stress [76,77]. Liu et al. (2024) mitigated the effects of H2O2 RPE degeneration by pretreating ARPE-19 cell monolayer models with lutein over a 24 h period [78]. Treatment with lutein showed diminished oxidative stress and a reduction in pro-inflammatory cytokines in the 3DP model [78].

### 6.4. Applications in Drug Development

The 3DP prototypes are increasingly favored for modeling treatment applications in retinal disease [79]. Beharry et al. (2018) exemplified the superiority of 3DP constructs by developing a comparison experiment between the application of a topical retinopathy of prematurity treatment on a co-culture of human retinal endothelial cells and human retinal astrocytes [80]. The 3D construct exhibited less oxidative stress and variability in drug response when compared with a 2D co-culture on media [80]. Similarly, Kim et al. (2022) advanced a model for RPE oxidative stress from cigarette smoke into a drug-testing platform by successfully applying antioxidants that suppressed further oxidative damage [72]. Direct drug application in 3DP models is currently limited due to an inability for models to fully biomimic the complexity of native retinal tissue. This is anticipated to improve with technological advancement.

### 6.5. Integrative Approaches to Retinal Tissue Engineering

Three-dimensional bioprinting of retinal tissues does not exist in a silo; the literature demonstrates relationships with microfluidics, organ-on-a-chip, and advanced imaging techniques to improve modeling. Microfluidic chips are devices that enable analysis of samples and chemicals on a very small scale by manipulating the physical and chemical properties of liquids and gases [81]. Microfluidic chips have been printed using similar extrusion and inkjet printing techniques to retinal tissue engineering [81]. Future applications could combine 3D-printed microfluidic chips to further investigate the behavior of retinal cells and to aid in cell processing. Sun et al. (2023) advanced a similar chip technology to cultivate retinal organoids (ROs) on a 3D-printed polydimethylsiloxane (PDMS) microwell platform [82]. This platform confines iPSC-derived ROs to individual microcavities while sharing the same medium and environment, preventing fusion and supporting long-term culture with fewer apoptotic cells [82]. The PDMS microwell platform using 3D bioprinting improves the efficiency and uniformity of ROs, potentially enhancing in vitro retinal organogenesis and standardization [82]. While it promotes RO maturation without the need for BMP4 or Matrigel, challenges remain in guiding the spatial arrangement of photoreceptor cells for transplantation. The gap in guiding the spatial arrangement of retinal cells was addressed in Kador et al. (2016)’s work with RGCs. An RGC’s cellular position is determined by guidance molecules secreted in fetal development; this process is difficult to replicate later in life even with transplanted RGCs [83]. Kador et al. combined radial electrospun cell transplantation scaffolds with traditional inkjet bioprinting to precisely position RGCs on a scaffold surface [83]. Future applications may apply a similar scaffolding process for guiding other non-regenerative cells elsewhere in the retina.

## 7. Barriers to Clinical Translation and Future Perspectives

### 7.1. Steps Toward Clinical Translation: From Bench to Bedside

Three-dimensional printing is a consistently reproducible, precise process that generates printed constructs faster than traditional bench work developments, closing the gap between experimental work and clinical application. Retinal constructs should emulate native tissue biomechanics accurately. Therefore, individual immunogenicity should be considered in the implantation of retinal constructs. Ocular immune privilege prevents major inflammation in the tissue microenvironment of the eye due to a complex physical barrier composed of tight junctions, specialized transmembrane proteins, the downregulation of various immune markers, and soluble negative immune regulators [89]. However, given the unknown nature of printed constructs in vivo, it is necessary to consider the possibility of requiring immunosuppressants to prevent undue inflammation and cellular damage of the construct. Suppressants should be tested in constructs and animal models to ensure tolerability prior to application in humans. Immunogenicity can be countered by seeding scaffolds with HLA-matched allogeneic iPSC-derived retinal progenitor cells [90,91].

The use of progenitor cells also generates a risk for tumorigenesis and teratoma formation from contamination and cellular reprogramming in the retina after a scaffold is placed. A recent study showed prevention of tumorigenesis in iPSC RPE cells in animal rejection models by culturing iPSCs without the traditional use of viral vectors and synthetic RNA products [92].

### 7.2. Potential Clinical Applications and Implications

Functional retinal constructs have several clinical applications (refer to Table 6). Retinal vascular diseases are common in the general population and a driver of vision impairment and blindness, requiring repeated intravitreal injections for management [93]. This can present multiple social and economic barriers for complex patients. Won et al. (2020) developed a drug-loaded rod to deliver two types of known drugs, bevacizumab (BEV) and dexamethasone (DEX), from a singular implant [94]. The rod is injected intravitreally using a small needle, a lesser invasive procedure compared with traditional regular injection [94]. The process of rod-enabled drug delivery is summarized in Figure 9. Animal testing of the rod showed a greater efficacy in reducing inflammation and providing long-term suppression of neovascularization compared with traditional BEV/DEX injections [94]. Novel retinal therapies are also applicable to retinal degenerative disease. Thompson and collaborators (2019) generated a 3D model using two-photon polymerization of polycaprolactone (PCL) to achieve optimal shape fidelity for retinal grafting [95]. The optimized PCL scaffold was successfully seeded with human iPSC-derived retinal progenitor cells. Subsequent implantation of the scaffold in a porcine retinitis pigmentosa model showed no adverse effects one month post-implantation [95].

Retinal tissue engineering enhances traditional 2D imaging modalities, supporting the visualization of deep vascular networks and surgically relevant vitreoretinal sites. Recent advancements in optical coherence tomography (OCT), including enhanced depth imaging optical coherence tomography (EDI-OCT) and swept source OCT (SSOCT) provide clinicians detailed views of the choroidal layers [96]. However, these imaging modalities are static and reduce clinician ability to view relevant features such as depth and spatio-anatomical localization [96]. Maloca et al. (2019) describes the novel use of SSOCT data to 3D print 13 individual patient choroidal vessels and pigmented choroidal tumors [97]. These models provide clinicians with pertinent information in assessing choroidal vasculature in the 3D plane and can provide insight in determining effects of choroidal inflammatory disease. OCT data have also been combined with stereolithography to recreate a patient with a rare pathological epiretinal membrane (ERM), supporting the learning of vitreoretinal surgical trainees in developing surgical plans for ERM removal [97].

Using 3D printing technology to create anatomically accurate models can be an immensely useful tool for patient education and surgical training. Few studies have previously explored this application on retina. Yap et al. (2017) first used OCT to print a retina model affected by age-related macular degeneration to educate patients [98]. After transforming OCT images into printable data, removing redundancies and noise, and scaling the model, it took 30 min to print out an entirely usable model [98].

In 2018, Choi et al. used OCT and stereolithography to create 3D models of macular pucker—the first report to apply 3DP with individualized patient imaging to visualize diseased retinal surfaces [97]. This customizability permits surgeons to understand the unique surface and shape of the epiretinal membrane for pre-operative surgical planning. Maloca et al. (2019) expanded upon their techniques to create a more detailed macula model, capturing more intricate features such as the foveolar avascular zone (FAZ) and vascular networks [99].

To evaluate clinician opinion, Pugalendhi et al. (2021) surveyed 69 ophthalmologists to compare previously existing models and their 3DP-generated model [100]. The 3DP model was superior in several aspects: on a technical level, it improved the viewing area and viewing angle. On a functional level, it improved the comfort levels of physicians. Overall, this modality provided a low-cost, reusable, and straightforward solution to improving surgical skills, confidence, and reducing errors.

### 7.3. Limitations of Current Review

While this review highlights significant and recent advances in 3D bioprinting for retinal tissue engineering, some limitations remain. This article did not explore the regulatory, ethical, or logistical barriers to clinical translation. Nor did it discuss the long-term viability, durability or functional integration of bioprinted retinal constructs in vivo due to a paucity of available data. Research addressing these gaps will be crucial to advancing 3DP toward clinical application.

### 7.4. Future Research Directions

The development of novel bioinks and evolving biomaterials have significantly increased cellular viability in retinal tissue engineering. Future projects should endeavor to better mimic the natural extracellular matrix of the retina in complexity and attempt to replicate the cellular diversity of retinal layers. Bioinks and biomaterials will require enhanced work in supporting long-term cellular viability as we look to generate in vivo therapies for transplantation and grafting. Incorporating growth factors, appropriate guiding molecules for cellular orientation, and supporting vascularization will lead to more robust and viable retinal constructs.

These improvements will move the field toward generating a multilayer retinal model with the ultimate goal of creating a fully functioning retina. This progression will usher in a promising future for regenerative medicine, with the potential to transform the landscape of retinal therapies and vision restoration.

## 8. Conclusions

In summary, 3D bioprinting has emerged as a powerful tool with the potential to recreate the complex architecture and functionality of retinal tissue, offering new avenues for research, drug testing, and, eventually, clinical applications. Although significant strides have been made in refining bioinks, optimizing scaffold materials, and enhancing printing techniques, several challenges remain. Achieving the precise cellular alignment, functional integration, and long-term viability necessary for retinal constructs is complex and requires further technological innovation. Additionally, issues such as immune compatibility and regulatory approval present obstacles that must be carefully addressed to transition these bioprinted constructs from laboratory models to clinical trials.

The future of 3D bioprinting in retinal tissue engineering is promising yet calls for measured optimism. Continuous improvements in bioink formulations, scaffold designs, and bioprinting precision will be essential to advance these models toward functional, transplantable retinal tissues. Ultimately, the success of translating these advances into clinical settings relies on the seamless collaboration between biomedical engineers, scientists, and ophthalmologists. Through sustained, interdisciplinary efforts, the path to restoring vision through bioprinted retinal constructs may one day become a reality, offering new treatment modalities for patients suffering from irreversible retinal diseases.

## Figures and Tables

**Figure 1 biomimetics-09-00733-f001:**
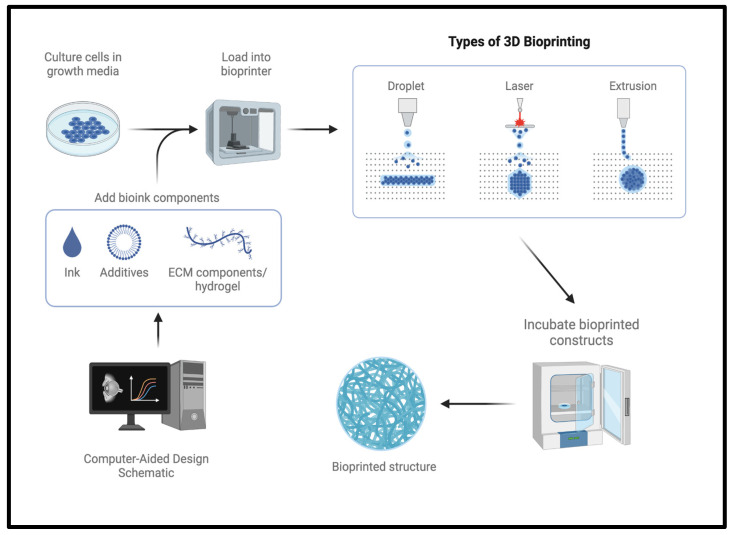
Three-dimensional bioprinting process and types of bioprinting. Created in BioRender.

**Figure 2 biomimetics-09-00733-f002:**
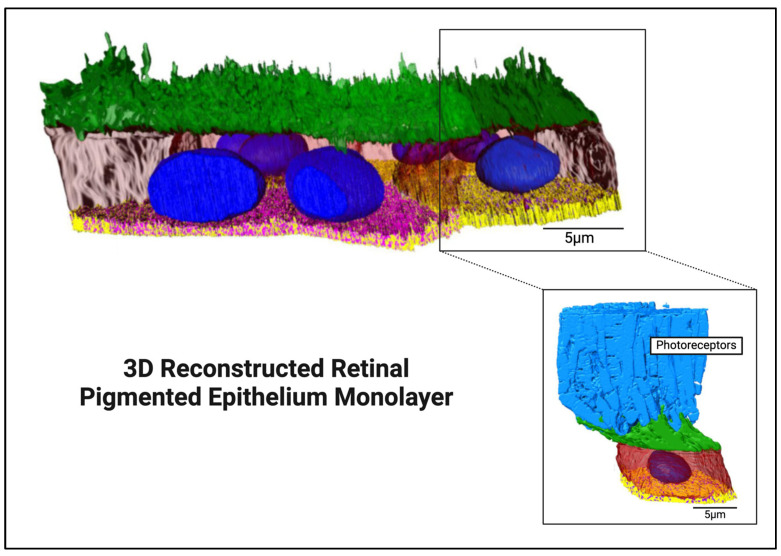
Keeling et al. [26] created reconstructed images of mice RPE 3D architecture (lateral view) showing apical microvilli (green) and nuclei (blue) with transparent cytoplasm allowing visualization of the convoluted basolateral Bruch’s membrane (yellow) with sub-RPE spaces (purple) and photoreceptors (light blue). Created in BioRender.

**Figure 3 biomimetics-09-00733-f003:**
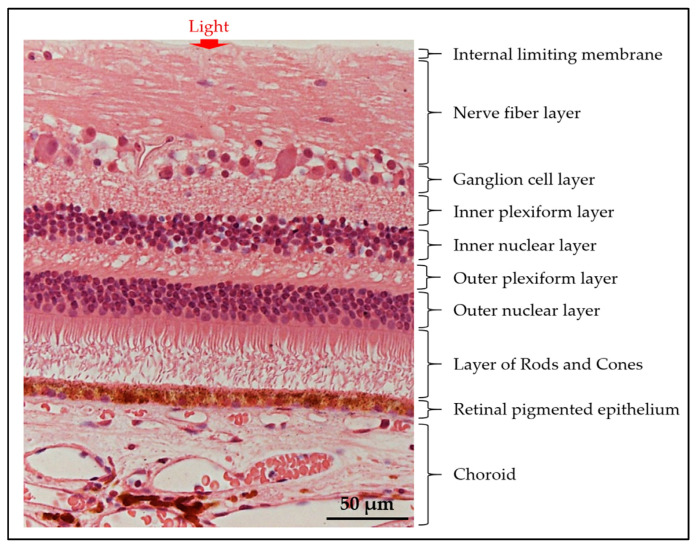
Deep to the outer pigmented aspect of the retina is the nine layers within the inner neural layer of the retina. The retina is located between the vitreous body and choroid [27]. Copyright certificate is CC by 3.0 license.

**Figure 4 biomimetics-09-00733-f004:**
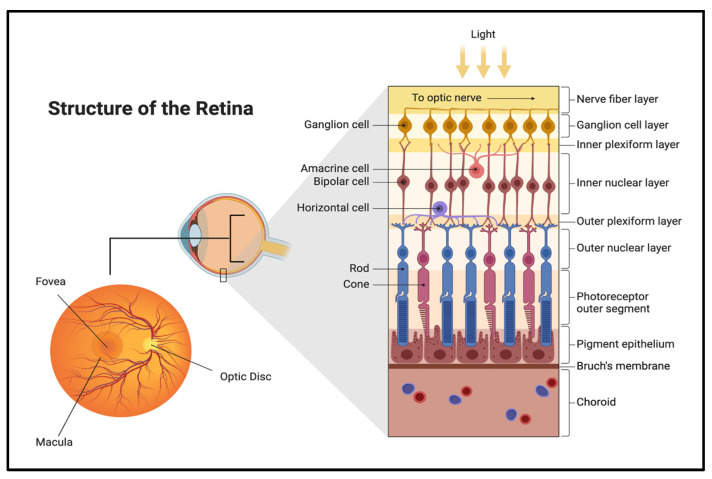
Retina structures cartoonized. Note: not all retinal layers are depicted in this figure. Created in BioRender.

**Figure 5 biomimetics-09-00733-f005:**
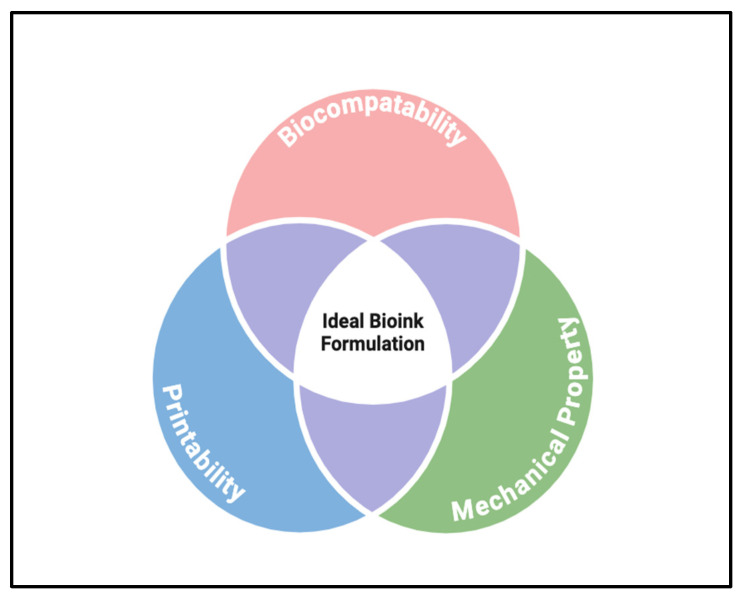
Diagrammatic representation of the major requirements for a successful bioink. Created in BioRender.

**Figure 6 biomimetics-09-00733-f006:**
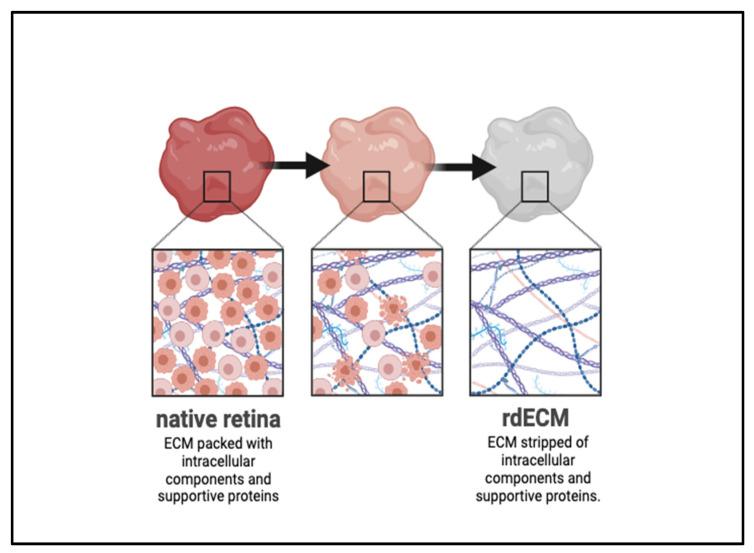
Cartoonized rendering of the decellularization process for the development of decellularized ECM (dECM) biomaterial. The progressive loss of colour in this figure represents the loss of intracellular components in the decellularization process. The native retina tissue for which the ECM is derived is rendered in red, emblematic of the complex protein structures and intracellular environment supporting the native ECM. The final dECM product is rendered in gray, stripped of the native supportive proteins and growth-promoting intracellular environment. Created in BioRender.

**Figure 7 biomimetics-09-00733-f007:**
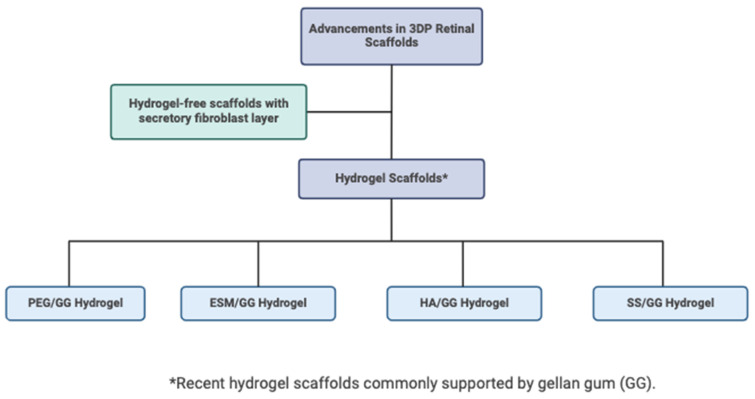
Flow chart summarizing recent advancements of scaffold engineering in 3D retinal bioprinting. Many scaffolds are made with gellan gum (GG) as a base for its improved strength during the printing process. Created in Biorender.

**Figure 8 biomimetics-09-00733-f008:**
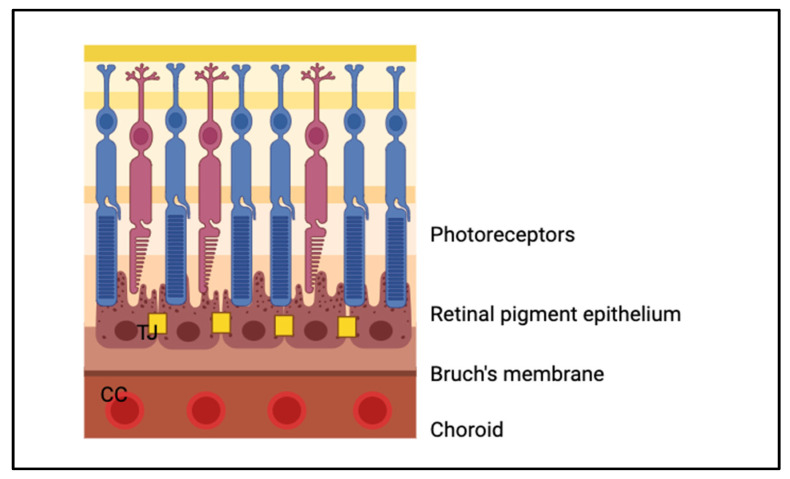
Schematic representation of the oBRB. CC = choriocapillaris; TJ = tight junction. Created in Biorender.

**Figure 9 biomimetics-09-00733-f009:**
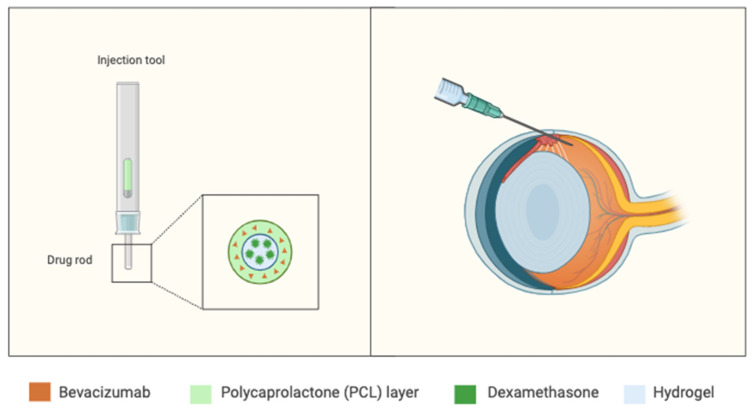
Graphical representation of the drug-loaded combined bevacizumab/dexamethasone rod invention [94]. Created in Biorender.

**Table 1 biomimetics-09-00733-t001:** Comparison of the three main Types of 3D bioprinting [19,20].

Criteria	Extrusion-Based	Droplet-Based	Laser-Based
Mechanism	Continuous bioink deposition through a nozzle	Controlled bioink droplets on a substrate	Laser pulses transfer bioink droplets
Resolution	Low to medium (100–500 µm)	Medium to high (10–50 µm)	Very high (1–10 µm)
Speed	Moderate	High	Moderate to low
Material Compatibility	Viscous bioinks	Low viscosity bioink	Broad, including delicate materials
Advantages	Versatile, scalable, cost-effective	High resolution, fast droplet generation	Extremely precise, minimal bioink waste
Disadvantages	Lower resolution, possible cell damage	Limited material choice, potential clogging	Expensive, complex, slower
Typical Applications	Large constructs (e.g., bone, vascular tissues)	High-precision models (e.g., skin, microfluidics)	Microscale tissue engineering, cell patterning

**Table 2 biomimetics-09-00733-t002:** Main cellular elements of the human retina.

Cell Type	Location	Function	Reference
Bipolar	Inner Nuclear Layer	Transmit signals from photoreceptors to ganglion cells.	[3]
Horizontal		Integrate input from multiple photoreceptors to enhance contrast and visual acuity.	
Amacrine		Modulate signal transmission between bipolar and ganglion cells. Also involved in temporal processing.	
Ganglion	Ganglionic Layer	Transmit visual information from the retina to the brain via the optic nerve.	
Photoreceptors	Rod and Cone Layer	Capture light and convert it into electrical signals for neural processing.	
Glial	Various Locations	Include Müller cells, astrocytes, and microglia which serve critical support and maintenance functions.	

**Table 3 biomimetics-09-00733-t003:** Main acellular elements of the human retina.

Biomolecule	Location in Retina	Function	References
Elastin	Bruch’s membrane	Elasticity, structural support, maintain flexibility of retinal layers	[3,19]
Collagen (Type I, IV)	Bruch’s membrane	Tensile strength, structural integrity, forms scaffold for cellular attachment.	
Laminin	Bruch’s membrane	Facilitate cell adhesion, differentiation, migration, critical for maintaining retina architecture.	
Fibronectin	Extracellular matrix	Wound healing, cell adhesion, and migration within the retinal tissue.	
Proteoglycans	Extracellular matrix	Help regulate cell behavior, contribute to hydration and spacing of retinal tissue.	
Perlecan	Bruch’s membrane	A heparan sulfate proteoglycan that regulates growth factors, provides structural support, and maintains the integrity of the basement membrane.	
Integrins	Retinal cells and Bruch’s membrane	Facilitate cell adhesion to the extracellular matrix, play a critical role in cellular communication, and mediate signal transduction.	

**Table 4 biomimetics-09-00733-t004:** Requirements for bioinks.

**Requirement**	**Determinants**	**References**
Biocompatibility	- Carrier materials (hydrogels vs. cell-laden biomaterial) - Hydrogel considerations: natural vs. synthetic - Cellular viability and differentiation - Cytotoxicity to local microenvironment - Ability to promote cell growth	[2,28,33,34,35]
Printability	- Viscosity - Viscoelasticity - Elastic recovery - Shear stress and shear thinning procedures	[32,36,37,38]
Mechanical Property	- Swelling degree in hydrogels - Physical and chemical cross-linking - Molecular weight - Gelation kinetics - Stiffness	[37,39,40]

**Table 5 biomimetics-09-00733-t005:** Applications of 3D bioprinting in retinal tissue engineering.

Requirement	Key Features		References
Optimization of Retinal Tissue Engineering	Cell-specific replication	- Neural cells (RGCs, glia) - Photoreceptors	[60,61,62]
	Hydrogel scaffolds	- Additives to a gellan gum carrier material to improve stability - Polyethylene glycol and hyaluronic acid are better for intravitreal injection and cell growth - Hydrogel-free alternatives (living biopaper) - Variety of printing methods	[63,64,65,66,67,68,69,70]
	Complex structure	- Vascularization potential post-printing - Multilayered (2+) retinal constructs - Introduction of RPE spheroids for cell regeneration	[41,71]
Retinal Disease Models	- RPE degradation from oxidative stress - Viral disruption of oBRB - AMD modeling		[72,73,74,75,76,77,78]
Drug Development	- Enhancement of retinopathy of prematurity treatment with 3DP co-culture - Antioxidants to reduce RPE oxidative damage		[72,79,80]
Integration with Secondary Technologies	- Microfluidic chip - Retinal organoids - Replicating native RGC positioning with radial electrospun scaffolds		[81,82,83]

**Table 6 biomimetics-09-00733-t006:** Clinical translations.

Retina Clinical Application	Description	Clinical Impact	References
Therapeutics	Drug-loaded rod alternative to intravitreal injection in vascular disease	Reduced economic burden for patients Long-term angiogenesis suppression	[94]
	Grafts for retinitis pigmentosa	Potential to reverse vision loss	[95]
Imaging	OCT data for 3DP of the choroid	Clinician insight into choroidal inflammatory disorders that affect the retina	[96]
Surgical Training	OCT data for 3DP of an AMD model	Patient education on disease and empowerment	[97,98]
	OCT data for 3DP of the macula	Improved understanding of 3D structures for medical learners Low-cost Reusable Error-reducing	[99,100]

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
