# Peer review of "Three-Dimensional Bioprinting for Retinal Tissue Engineering"

_biomimetics, 2024, doi:10.3390/biomimetics9120733_

Round 1

Reviewer 1 Report

Comments and Suggestions for Authors

        The manuscript by Wu et al. have reviewed the recent innovations and the applications of 3D bioprinting in retinal tissue engineering. The authors discussed different methods used for 3D printing and how they can mimic the anatomy and function of the retina at tissue level. The authors also reviewed the progress and limitations of human-on-chip platforms, which is of great  importance to the field. I like the “Barriers to Clinical Translation and Future Perspectives” part, it is challenging for now to fully mimic retinal diseases for translational medicine application. The review is well written with informative tables and discussions.  

        I have one minor concern: the authors discussed the cells and materials (bio-ink) used for retina bioprinting, another challenge is how to make those different types of cells live happy and longer in the printed tissue. It would be great if the authors could discuss cell culture media optimization and vascularization with bio-printed endothelial cell in section 6.2.

Author Response

Dear Reviewer,

Thank you for your thoughtful and positive feedback on our manuscript. We deeply appreciate your recognition of the manuscript's contributions to the field of retinal tissue engineering, particularly your remarks on the "Barriers to Clinical Translation and Future Perspectives" section. Your kind words encourage us greatly.

We also sincerely thank you for highlighting the importance of addressing cell culture media optimization and vascularization in the context of 3D bioprinting. We have incorporated your suggestion by expanding the discussion in Section 6.2.3 to include the latest insights into cell culture media optimization for supporting cell viability and functionality, as well as the potential of bioprinted endothelial cells to promote vascularization in retinal constructs.

We believe these updates strengthen the manuscript and provide a more comprehensive perspective on the challenges and advancements in the field.

Once again, thank you for your constructive feedback and for the opportunity to improve our work. We hope the revised manuscript meets your expectations and are happy to address any additional concerns.

Best regards,

Reviewer 2 Report

Comments and Suggestions for Authors

This was a good topic in 3D bioprinting area, and this paper was well-organized. For attracting readers in related areas, I suggest the authors should further improve the manuscript. There were many review papers elaborating bioprinting principles, different bioprinting techniques, and retinal tissue engineering, and bioinks preparation rules, which were similar to some of content in this paper, until section 6 (page 13), so the authors should focus on recent reports and description after section 6. especially some recent advances should be clearly elaborated with some figures including detailed information. 

Author Response

We sincerely thank the reviewer for their thoughtful feedback and for recognizing the value of our topic in 3D bioprinting. We appreciate the detailed suggestions for improving the manuscript, which have helped us enhance its quality and relevance to the field. Below, we address each of the reviewer’s comments:

  1. General comment on overlap with previous reviews:
    We understand the reviewer’s concern regarding the overlap of certain sections with existing review articles. To address this, we have revised the manuscript to focus more extensively on recent advancements and innovative developments after section 6. Specifically, we have incorporated additional discussions of recent reports and highlighted key breakthroughs in sections 6 and 7.

  2. Suggestion to elaborate on recent advances with detailed figures:
    Following the reviewer’s recommendation, we have included two new figures in the manuscript to provide detailed visual representations of the discussed advancements:

    • Figure 1 (Section 6.2.2): A flowchart summarizing recent advancements in scaffold engineering for 3D retinal bioprinting, highlighting the use of gellan gum as a base material for its improved strength during printing.
    • Figure 2 (Section 7.2): A graphical representation of the drug-loaded combined bevacizumab/dexamethasone rod invention by Won et al. (2020).

    These figures, created with Biorender, aim to enhance clarity and attract readers’ attention to the recent innovations discussed in our manuscript.

We have carefully revised the manuscript to ensure these additions are seamlessly integrated and provide meaningful insights to the readers. Thank you once again for your valuable suggestions and for helping us improve the quality and impact of our work.

Sincerely,

Reviewer 3 Report

Comments and Suggestions for Authors

The presented work is a review of 3D bioprinting of retinal tissue. This work broadly covers the path toward clinical applications, focusing on overcoming immunogenicity, ensuring durability, and refining biomimetic strategies for potential breakthroughs in vision restoration. The manuscript is well-written and organized. The figures are apt and appropriate and support the study. Retinal tissue engineering needs rapid advancements, and this work highlights the key issues and innovations that are much needed for retinal-engineered tissues. However, some of the sections need attention. 

1.        In the introduction section specific gaps in this research need to be defined clearly.

2.        The printing methods and processes such as extrusion, laser, and droplet methods were explained properly and provided in-depth information.

3.        Bioink formulations were addressed thoroughly. However, adding the mechanistic role of bioinks in retinal cell functionality and viability will strengthen the section.

4.        Extracellular matrix will be crucial for structural integrity and support and providing a boundary between the layers. Hence having a separate discussion on ECM will be highly impactful.   

5.        Add the mechanical properties and numerical ranges that are essential for the printed tissue.

6.        Also, it will be interesting to discuss some mechanical factors and associated challenges that impact vascularization.

7.        Tables and Figures are well-labeled and clear. The barriers and clinical translational section are written exceptionally well. 

8.        The Zika virus model cannot be generalized in its current state. It lacks immunological diversity.  Adding a discussion about ocular immune responses and immunological responses will be advantageous for the readers and may enhance the relevance of the model. 

9.        Why are the blood-retina barrier and its clinical relevance not discussed separately?

10.    I strongly recommend adding a limitation of this study.  

Author Response

We sincerely thank the reviewer for their thoughtful and constructive feedback, which has significantly improved the quality of our manuscript. Below, we address each of the points raised and outline the corresponding revisions made.

  1. In the introduction section, specific gaps in this research need to be defined clearly.
    Response:
    Thank you for highlighting this important aspect. The introduction section has been revised to clearly define specific gaps in the current research landscape. We have included examples to illustrate the pressing challenges and areas where advancements are most needed.

  2. Bioink formulations were addressed thoroughly. However, adding the mechanistic role of bioinks in retinal cell functionality and viability will strengthen the section.
    Response:
    We appreciate your suggestion. Section 4.1 has been expanded to include a detailed discussion on the mechanistic roles of bioinks in supporting retinal cell functionality and viability.

  3. Extracellular matrix (ECM) will be crucial for structural integrity and support, and providing a boundary between the layers. Hence, having a separate discussion on ECM will be highly impactful.
    Response:
    Thank you for this valuable insight. A separate discussion on the extracellular matrix and its critical role in retinal tissue engineering has been added to Section 4.1, emphasizing its structural and functional importance.

  4. Add the mechanical properties and numerical ranges that are essential for the printed tissue.
    Response:
    We carefully considered this recommendation and discussed it among the co-authors. While we recognize the value of including detailed mechanical properties and numerical ranges, we believe that incorporating such technical data may extend beyond the exploratory scope of our review. Our focus remains on overarching trends and potential applications. This choice allows the manuscript to engage a broader interdisciplinary audience. We plan to address this in future technical studies and hope this explanation aligns with the reviewer’s perspective.

  5. It will be interesting to discuss some mechanical factors and associated challenges that impact vascularization.
    Response:
    Thank you for this excellent point. Discussions on the mechanical factors affecting vascularization in retina-specific models were already included in Sections 3.2, 5.1, and 6.2.3. To enhance clarity, we have refined these sections to better highlight the challenges and solutions related to vascularization in 3D bioprinted retinal tissue.

  6. The Zika virus model cannot be generalized in its current state. It lacks immunological diversity. Adding a discussion about ocular immune responses and immunological responses will be advantageous for the readers and may enhance the relevance of the model.
    Response:
    Thank you for this thoughtful suggestion. We have revised the relevant section to include a discussion on ocular immune responses and the immunological diversity required to enhance the applicability of the Zika virus model.

  7. Why are the blood-retina barrier (BRB) and its clinical relevance not discussed separately?
    Response:
    To maintain a smooth narrative, we integrated the discussion of the BRB within the manuscript. Based on your feedback, we have enhanced Section 3.1 to include additional information about the biological role and clinical relevance of the blood-retina barrier. We believe this addition addresses your concern while preserving the manuscript's flow.

  8. I strongly recommend adding a limitation to this study.
    Response:
    Thank you for the recommendation. We have added a dedicated section on the limitations of this review in Section 7.3, providing a critical evaluation of its scope and outlining areas for future research.

We sincerely thank the reviewer once again for their constructive comments and hope that the revised manuscript meets their expectations. Please do not hesitate to provide further feedback or suggestions.

Sincerely,

Round 2

Reviewer 2 Report

Comments and Suggestions for Authors

The paper can be accepted in current form.